# Potent Inhibition of Chikungunya Virus Entry by a Pyrazole–Benzene Derivative: A Computational Study Targeting the E1–E2 Glycoprotein Complex

**DOI:** 10.3390/ijms26136480

**Published:** 2025-07-05

**Authors:** Md. Mohibur Rahman, Md. Belayet Hasan Limon, Tanvir Ahmed Saikat, Poulomi Saha, Abdul Hadi Nahid, Mohammad Mamun Alam, Mohammed Ziaur Rahman

**Affiliations:** 1One Health Laboratory, Infectious Diseases Division, International Centre for Diarrhoeal Disease Research, Mohakhali, Dhaka 1212, Bangladesh; mohibur.rahman@icddrb.org (M.M.R.); tanvir.saikat@icddrb.org (T.A.S.); poulomi.saha@icddrb.org (P.S.); abdul.nahid@icddrb.org (A.H.N.); mzrahman@icddrb.org (M.Z.R.); 2Department of Microbiology and Immunology, University of Louisville, Louisville, KY 40202, USA; bhasanlimon@gmail.com

**Keywords:** Chikungunya, E1–E2 envelope glycoprotein, virtual screening, molecular docking, molecular dynamics simulations, entry inhibitors

## Abstract

The Chikungunya virus (CHIKV) continues to pose a significant global health challenge due to the absence of effective antiviral treatments and limited vaccine availability. This study employed a comprehensive in silico workflow, incorporating high-throughput virtual screening, binding free-energy calculations, ADMET (absorption, distribution, metabolism, excretion, and toxicity) analysis, and 200 ns molecular dynamics (MD) simulations, to identify new inhibitors targeting the E1–E2 glycoprotein complex, crucial for CHIKV entry and membrane fusion. Four promising candidates were identified from a library of 20,000 compounds, with CID 136801451 showing the most potent binding (docking score: −10.227; ΔG_bind: −51.53 kcal/mol). The top four compounds exhibited favorable ADMET profiles, meeting nearly all criteria. MD simulations confirmed stable binding and strong interactions between CID 136801451 and the E1–E2 complex, evidenced by consistently low RMSD values. These findings highlight CID 136801451 as a promising CHIKV entry inhibitor, warranting further in vitro and in vivo evaluation to advance the development of effective anti-CHIKV therapeutics.

## 1. Introduction

Chikungunya virus (CHIKV), an alphavirus in the family of Togaviridae, is transmitted to humans by *Aedes* mosquitoes, primarily by *Aedes aegypti* and *Aedes albopictus* [1]. The first documented Chikungunya outbreak in Tanzania involved a few hundred cases. Since then, CHIKV has spread to over 110 countries [2]. CHIKV is widespread in tropical and subtropical regions, putting over 75% of the global population at risk. Currently, 1.3 to 2.7 billion people live in areas vulnerable to the Chikungunya virus [3,4]. From December 2013 to June 2023, 3.7 million Chikungunya cases, including suspected and laboratory-confirmed cases, were documented in 50 American countries or territories [5].

The acute phase of Chikungunya virus infection causes sudden fever, severe joint pain, headache, muscle discomfort, joint swelling, fatigue, and rash. On rare occasions, the joint pain linked to Chikungunya virus infection might progress into a subacute or chronic state, persisting for several months or even years [6,7]. With the global prevalence of the *Aedes* mosquito vectors and high viral titer in the blood of infected patients, exceeding 5 log_10_ plaque-forming units per mL of blood, CHIKV outbreaks exhibit exceptionally high attack rates [8]. Despite the widespread prevalence and impact, the current state of treatment and prevention for Chikungunya remains a critical concern.

Currently, there is no targeted antiviral therapy or widely accessible vaccination for Chikungunya infection. Management focuses on alleviating symptoms using analgesics and antipyretics [2]. In November 2023, the FDA approved the first CHIKV vaccine, Ixchiq, a live attenuated vaccine. The U.S. Advisory Committee on Immunization Practices (ACIP) recommends it only for travelers to high-risk areas and laboratory workers [6]. However, despite the approval, the vaccine is not widely available. Moreover, no vaccine is 100% effective, meaning breakthrough infections can still occur. Antiviral therapeutics would be essential for managing the disease [9].

Chikungunya virus (CHIKV) is a spherical virus with viral envelope proteins on its surface and a single-stranded, positive-sense RNA genome of approximately 11.8 kilobases. CHIKV utilizes MXRA8 (Matrix Remodeling Associated 8) as a functional receptor [10]. The virus enters host cells when its E2 glycoprotein binds to the hMXRA8 receptor, initiating receptor-mediated endocytosis. The CHIKV E1–E2 heterodimer dissociates within acidic endosomes, and E1 rearranges into homotrimers, exposing the fusion loop. This fusion loop facilitates the merging of the viral envelope with the endosomal membrane, enabling viral entry and subsequent replication inside the host cell [11,12].

Recent studies show that small molecules disrupting the interaction between E2 glycoprotein and the hMXRA8 receptor could be a potential antiviral treatment [10,13]. Several computational and experimental studies have explored methods to interfere with the early stages of CHIKV infection. Epigallocatechin-3-gallate (EGCG) from green tea has been found to prevent CHIKV attachment, significantly reducing infection rates [14]. Other drugs, such as arbidol, chloroquine, curcumin, suramin, and micafungin, have also been shown to disrupt the process of CHIKV cell entry [15,16].

Despite continuous research efforts, developing targeted antiviral medications for CHIKV remains elusive. Many antiviral compounds tested against CHIKV have shown toxicity or side effects, which can limit their use in humans [17]. This highlights the need for more research into the safety profiles of potential antiviral medications. In this study, we utilized high-throughput virtual screening to analyze several small molecule libraries sourced from various databases by targeting a specific binding location on the E1–E2 dimer, focusing on the critical amino acid residues involved in the interaction with the hMXRA8 and membrane fusion. Specifically, our target site encompasses the hMXRA8-interacting region and the fusion loop of the CHIKV envelope glycoprotein. This dual-targeting strategy was selected to block CHIKV entry into host cells in two steps: first, by preventing the interaction between the envelope and the host cell receptor and second by inhibiting membrane fusion via the fusion loop. The investigation has yielded a notable outcome by identifying a novel benzene derivative incorporating a pyrazole group (CID 136801451), demonstrating specific anti-CHIKV properties. This could be a crucial step in developing potential therapeutics against CHIKV infection.

## 2. Results

### 2.1. Quantitative Evaluation of Ligand Bindings

The docking study results presented in Table 1 outline the calculated binding affinities of four distinct compounds, assessed by the Glide software’s (Schrödinger Release 2023-2) XP GScore (Extra Precision Glide Score) and docking score. The XP GScore and docking score are highly consistent, suggesting that the initial docking poses accurately reflect the energetically favorable conformations.

CID 136801451 stands out with the most significant binding free energy, as indicated by its docking score of −10.227 and suggesting a strong interaction with the target protein. Although, CID 25246271 shows a slightly more favorable XP GScore but a lower docking score compared to CID 136801451. CID 5496862 and CID 3894 showed docking scores of −9.362 and −9.169, respectively.

### 2.2. Interactions and Residue Contacts Between Protein and Ligands

The 2D interaction diagrams of the E1–E2 protein complex with the ligands are exhibited in Figure 1. For CID 136801451, multiple hydrogen bonds are noted with residues such as TRP89 (E1), ASN72 (E2), and SER30 (E2), with repeated interactions involving SER124 (E2) indicating a possible key anchoring point within the binding site, as shown in Figure 1A. The compound also forms hydrophobic contacts with residues, including MET70 (E2) and multiple cysteine residues, which likely contribute to the stabilization of the compound within the protein binding pocket. From Figure 1B, we can see that CID 25246271 forms a single hydrogen bond with ASN72 (E2) but has extensive hydrophobic interactions, including with amino acid residues like ILE121 (E2) and TYR69 (E2). CID 5496862 and CID 3894 also exhibit a diverse array of non-bonded interactions, including hydrogen bonds and hydrophobic contacts, with crucial residues such as ARG178 (E2) and HIS29 (E2), shown in Figure 1C and Figure 1D, respectively.

### 2.3. ADME/T Properties Analysis

The ADME (absorption, distribution, metabolism, and excretion) and toxicity profiles of compounds identified as potential inhibitors of the Chikungunya virus were systematically evaluated and are presented in Table 2. CID 136801451 stands out as the most promising compound, exhibiting strong oral absorption, favorable permeability (QPPCaco: 53.22), and minimal central nervous system (CNS) activity, making it an ideal candidate for further development. It also demonstrates a good balance of lipophilicity and solubility, with a QPlogPC16 of 13.309 and a QPlogS of −4.176, although it raises slight concern for cardiotoxicity (QPlogHERG: −5.794). CID 5496862 shows favorable oral absorption and solubility, with a QPlogPC16 of 14.564 and a QPlogS of −4.849, but its permeability (QPPCaco: 17.129) and overall bioavailability are lower than CID 136801451. CID 3894 offers better skin permeability (QPlogKp: −3.670) and moderate metabolic stability (seven reactions), but its permeability and solubility are still lower compared to CID 136801451, making it less suitable for oral bioavailability. CID 25246271, however, shows poor performance across multiple parameters, including low solubility (QPlogS: −0.618), permeability (QPPCaco: 0.702), and human oral absorption (HOAP: 10.183), indicating it is less favorable for oral administration and overall efficacy. Despite these differences, all compounds comply with the Rule of Five, indicating they generally possess favorable drug-like properties.

### 2.4. Molecular Dynamics Simulation

#### 2.4.1. RMSD (Root-Mean-Square Deviation) Analysis

In Figure 2, the stability of the Chikungunya virus (CHIKV) envelope protein in complex with distinct ligand candidates was analyzed through root-mean-square deviation (RMSD) calculations over a 200 ns molecular dynamics (MD) simulation. The left axis of the provided graphs represents the RMSD range of protein–ligand complexes.

Here, the native protein (3N42) is used as a reference for the stability of the protein, which shows an RMSD of around 3.4 Å. CID 136801451 demonstrates the stability of the complex, as the RMSD consistently stayed around 4 Å. This indicates that the ligand maintains a stable interaction with the target protein, with minimal deviation from its reference conformation throughout the simulation period. CID 25246271 showed RMSD fluctuations with values rising above 5 Å, suggesting that the interaction between the ligand and the protein might be more flexible, which could lead to less stability in how the ligand fits within the protein’s binding site. The other CID 5496862 and 3894 also have higher RMSD values (>10 Å) and variability, indicating less conformational stability relative to CID 136801451.

#### 2.4.2. RMSF (Root-Mean-Square Fluctuation) Analysis

The root-mean-square fluctuation (RMSF) analysis elucidates the flexibility and dynamic behavior of protein residues during molecular dynamics simulations, as seen in Figure 3. Peaks on the RMSF plot correspond to regions of high flexibility, where the protein’s residues exhibit significant movement relative to their average structure.

In the presented plot, the native apoprotein (3N42) displays an RMSF of fewer than 3 Å, which is within the typical range for such proteins. As expected, the N-terminal and C-terminal regions demonstrate increased flexibility, a characteristic commonly observed in protein structures. In the complex with CID 136801451, the RMSF closely mirrors that of the apoprotein, with key ligand-contacting residues such as HIS26 (1.986 Å), CYS28 (1.112 Å), HIS29 (0.907 Å), ASP71 (1.111 Å), ASN72 (1.117 Å), HIS73 (0.984 Å), ARG119 (1.970 Å) of E2, and TRP89 (1.003 Å) of E1 showing minimal fluctuations. This indicates that the binding of CID 136801451 does not induce significant structural changes.

Conversely, in the complex with CID 25246271, elevated RMSF values were observed at several ligand-contacting residues, notably SER118 (3.505 Å) and ARG119 (3.363 Å) of E2, where RMSF values exceeded 3 Å. Similar to the effect seen with CID 25246271, the complex with CID 5496862 exhibited increased RMSF values at critical residues, suggesting that both CID 5496862 and CID 25246271 induce an induced-fit mechanism in the protein.

Furthermore, the complex with CID 3894 displayed the highest RMSF values, with particularly notable peaks around residues 200–300 and 600–700. This suggests that CID 3894 induces considerable conformational flexibility in the protein.

#### 2.4.3. Protein–Ligand Contact Mapping

The protein–ligand interaction histograms provide a quantified visual representation of the interaction types between the CHIKV envelope protein and various ligand compounds across the simulation trajectory. The interactions are categorized into hydrogen bonds (green bars), hydrophobic interactions (purple bars), ionic bonds (red bars), and water bridges (blue bars). The interaction fraction values are normalized to the duration of the simulation, where a value above 1.0 indicates multiple contact events at a single residue.

The protein–ligand contact map for the CHIKV envelope E1–E2 protein in complex with CID 136801451, as shown in Figure 4A, illustrates an extensive array of hydrogen bonds with SER30, MET70, ASN72, and SER124 of E2. Additionally, many hydrophobic contacts are present with HIS123 of E2 and TRP89 of E1, and a water bridge is present with HIS29. The contact map for CID 25246271 in Figure 4B shows fewer hydrogen bonds than the previous compound, primarily with ARG119 and, to some extent, with ASP71 of E2. Additionally, water bridges and hydrophobic bonds are present. The complex with CID 5496862 in Figure 4C demonstrates strong hydrogen bonds with ARG178 of E2 and GLU264 of E1, as well as some presence of hydrophobic bonds and salt bridges. The complex with CID 3894 in Figure 4D shows a prominent display of hydrogen bonds with ARG178 of E2 and GLY90 of E1. While hydrophobic interactions are evident with HIS29 of E2 and TRP89 of E1, many water bridges exist with HIS29, LEU241 of E2, and MET88, GLY91 of E1.

#### 2.4.4. Protein–Ligand Contact Timeline

The protein–ligand contact timelines provide valuable insights into the binding stability and interaction dynamics of the ligands with the CHIKV envelope E1–E2 protein. These contacts represent the various types of interactions observed during the simulation, including hydrogen bonds, hydrophobic interactions, ionic contacts, and water bridges. Figure 5 shows the protein–ligand contact timelines for the CHIKV envelope E1–E2 protein in complex with four different ligands (CID 136801451, CID 25246271, CID 5496862, and CID 3894) over a 200 ns molecular dynamics (MD) simulation. The y-axis represents the total number of contacts between the ligand and the protein, while the x-axis represents the simulation time in nanoseconds (ns).

CID 136801451 (Figure 5A) demonstrates the most stable and consistent interaction throughout the simulation. The contact numbers remain relatively constant (around 10) over time, indicating that this ligand maintains a strong and persistent binding with the CHIKV envelope protein throughout the entire simulation. In contrast, CID 25246271 (Figure 5B) exhibits a more transient and unstable binding pattern. The contact numbers fluctuate between 0 and 12, with several periods where the contact count drops to near zero. These fluctuations suggest that CID 25246271 fails to maintain consistent contact with the protein, likely forming weaker or less stable binding interactions.

CID 5496862 (Figure 5C) shows a similar fluctuating contact pattern to CID 25246271, with contact numbers ranging between 4 and 12. Although the contact numbers are higher compared to CID 25246271, they are still less stable and lower than those observed with CID 136801451. This indicates that CID 5496862 has low to moderate binding stability with the CHIKV envelope protein. CID 3894 (Figure 5D) initially shows few contacts with the protein, but after approximately 25 ns, the ligand maintains contact numbers ranging from 5 to 10 for the remainder of the simulation. This suggests that CID 3894 forms a moderately stable protein–ligand interaction after an initial period of weaker binding.

#### 2.4.5. MM/GBSA Computations from MD Trajectories

Post-simulation MM/GBSA calculations were performed using frames 1–500 from the MD simulation trajectories, totaling 500 conformations for each simulated complex. The average binding energies, along with their standard deviations, are presented in Table 3. MM/GBSA is a widely used and rigorous method for post-simulation binding free energy prediction because it accounts for factors such as protein flexibility, entropy, solvation, and polarizability—factors often overlooked in many docking protocols. Among the four CHIKV envelope protein–ligand complexes, CID 136801451 exhibits the strongest binding affinity, with a docking energy of −51.53 kcal/mol and a significantly improved MD simulation binding energy of −62.48 ± 9.06 kcal/mol, indicating strong and stable interactions. CID 3894 shows a docking energy of −24.18 kcal/mol and an enhanced MD energy of −38.08 ± 8.40 kcal/mol, suggesting it also stabilizes over time. CID 5496862 demonstrates moderate binding, with a docking energy of −20.06 kcal/mol and a stronger MD energy of −32.43 ± 11.62 kcal/mol, indicating an improvement in binding during the simulation. Conversely, CID 25246271 shows the weakest binding in both docking (−3.58 kcal/mol) and MD simulation (−8.19 ± 4.84 kcal/mol).

## 3. Discussion

Chikungunya fever has become a significant public health concern worldwide in tropical and subtropical regions. Currently, mosquito control and a limited distribution of vaccines are the preventive measures for the Chikungunya virus. However, mosquito vectors’ adaptability and widespread distribution make control efforts increasingly difficult, underscoring the urgent need for effective antiviral treatment. To address this critical gap, this study aims to identify and evaluate promising compounds that exhibit strong binding affinity, favorable pharmacokinetic properties, and stability in inhibiting the viral entry process, thereby contributing to the development of effective antiviral therapies for CHIKV infection to address the lack of therapeutic antiviral presence.

The discovery of inhibitors that block viral fusion and entry opens up new pathways for developing specific antiviral drugs. This study’s robust computational workflow, combining high-throughput virtual screening, molecular docking, binding free energy calculations (MM-GBSA), and molecular dynamics simulations, successfully identified a novel compound (CID 136801451) that exhibited a strong binding affinity and interaction with the CHIKV E1–E2 protein complex, mainly targeting key residues involved in viral entry and membrane fusion.

CID 136801451 exhibited the highest binding affinity, with a docking score of −10.227 and a ΔG bind value of −51.53 kcal/mol, reflecting strong interactions with key residues in the E1–E2 protein complex of CHIKV. ADME/Tox analysis revealed that CID 136801451 possesses favorable pharmacokinetic properties, including good oral absorption, metabolic stability, and balanced solubility and permeability. Although there were minor concerns regarding cardiotoxicity related to hERG inhibition, the compound remained within acceptable thresholds for most ADMET parameters.

Molecular dynamics simulations further validated the stability of CID 136801451 within the E1–E2 complex. Over the course of a 200-nanosecond simulation, the compound maintained a consistent root-mean-square deviation (RMSD) of approximately 4 Å, reflecting its stable binding. This low RMSD was supported by around 10 stable protein–ligand (P-L) contacts, as shown in Figure 5A. In contrast, CID 25246271 formed fewer P-L contacts (Figure 5B) and displayed greater fluctuations in RMSD, indicating lower stability. This observation was consistent with the post-simulation MM-GBSA analysis, where CID 25246271 exhibited the weakest binding, with a docking energy of −3.58 kcal/mol and a binding energy from the MD simulation trajectories of −8.19 ± 4.84 kcal/mol.

CID 5496862 initially maintained few P-L contacts (up to 75 ns), with several drops in contact numbers (Figure 5C), leading to an increase in RMSD, which reached up to 16 Å. However, after 75 ns, CID 5496862 stabilized, maintaining around 8 P-L contacts with the target protein, which resulted in reduced RMSD fluctuations and a stable RMSD value of approximately 13 Å for the remainder of the simulation. Similarly, CID 3894 exhibited a dramatic increase in RMSD, rising from 0 to 20 Å within the first 10 ns. During the initial 25 ns, the compound experienced significant RMSD fluctuations due to a low number of P-L contacts (Figure 5D). As the simulation progressed, CID 3894 transitioned from its initial reference position (obtained through protein–ligand docking in virtual screening) to a new conformation, forming more stable interactions, including hydrogen bonds, hydrophobic interactions, ionic bonds, and water bridges. This transition led to a more stable interaction with the CHIKV envelope protein, supported by reduced RMSD fluctuations after 25 ns.

Both CID 5496862 and CID 3894 initially exhibited high RMSD values due to the movement of the ligands from their reference positions to new conformations. This movement resulted in inefficient interactions with the target amino acid residues (Table 3) of the CHIKV E2–E1 protein complex, as shown in the protein–ligand contact maps (Figure 4). These inefficiencies indicated that CID 5496862 and CID 3894 were less effective as CHIKV entry inhibitors compared to CID 136801451. Furthermore, the MM/GBSA calculations from MD trajectories strongly support the candidacy of CID 136801451 as a potent CHIKV antiviral, with a docking energy of −51.53 kcal/mol and a significantly improved MD simulation binding energy of −62.48 ± 9.06 kcal/mol, highlighting its superior binding affinity compared to the other compounds studied.

Multiple studies [18,19,20] have emphasized the role of viral entry as a therapeutic target for combating the Chikungunya virus (CHIKV). Zhang et al. (2018) identified the hMXRA8 receptor as crucial for CHIKV entry, demonstrating that the viral E1–E2 glycoproteins interact with this host receptor to mediate attachment [10]. Furthermore, Battini et al. (2021) highlighted the significance of targeting the E1 fusion loop to inhibit viral membrane fusion [18]. Extending these insights, our research has identified compounds, particularly CID 136801451, that exhibit strong binding affinities to the E1–E2 protein complex, explicitly targeting critical residues such as TRP89 (E1) and ASN72 (E2), which are essential for viral attachment and fusion. This aligns with the findings of Song et al. (2019), which underscored the role of these residues in facilitating viral entry into host cells [12]. The interaction of CID 136801451 with both E1 and E2 envelope proteins suggests its potential to block two crucial steps in viral entry: the viral envelope–host receptor interaction and subsequent membrane fusion.

Small molecules that target the envelope proteins of virions can effectively inhibit the viral replication cycle at its earliest stage by binding extracellularly to the virus. De Wispelaere et al. (2018) identified a conserved pocket on the envelope proteins of several mosquito-borne flaviviruses, suggesting this region is a potential target for broad-spectrum antivirals with demonstrated efficacy in vitro [21]. Similarly, Helle et al. (2006) found that Cyanovirin-N (CV-N) inhibited hepatitis C virus (HCV) infectivity by binding to its heavily glycosylated envelope glycoproteins [22]. This interaction blocked the binding of the viral E2 protein to the host cell receptor CD81, underscoring the potential of targeting viral envelope proteins as a promising antiviral strategy.

As reported in the existing literature, several compounds have shown promising results in inhibiting CHIKV entry. Weber et al. (2015) demonstrated that epigallocatechin-3-gallate (EGCG), a compound found in green tea, effectively reduces CHIKV infection by preventing viral attachment to host cells [14]. Similarly, Battini et al. (2021) identified a small molecule, through structure-based virtual screening, that targets the E2–E1 glycoprotein complex, inhibiting viral internalization during CHIKV entry [18]. However, in our study, we focused on a unique region of the CHIKV E2–E1 glycoprotein. CID 136801451 exhibited higher docking scores and binding free energy, suggesting a more potent inhibition mechanism with robust and stable interactions at the viral entry point.

Some studies [23,24,25] have also identified suramin as an inhibitor of CHIKV entry, acting through interference with the E1–E2 glycoprotein complex. However, suramin’s high toxicity and limited specificity to CHIKV diminish its potential as a therapeutic candidate. Our study addresses these limitations by focusing on compounds with favorable ADMET profiles. The compounds show favorable polar surface areas, lipophilicity, and solubility for oral absorption and bioavailability. Specifically, CID 136801451 demonstrated excellent oral absorption and metabolic stability, making it a promising therapeutic option.

Additionally, molecular dynamics (MD) simulations validate the initial docking results by providing a dynamic perspective on the binding interactions [26]. MD simulations confirmed that CID 136801451 maintains stable binding over an extended period, supporting its capacity for sustained inhibition of viral entry. This stability has not been observed in compounds like EGCG, arbidol, suramin, or similar agents in previous studies, further highlighting the potential of CID 136801451.

The E1–E2 glycoprotein complex of the Chikungunya virus undergoes substantial conformational changes during the process of viral entry into host cells. Upon receptor binding and a subsequent drop in pH, the complex shifts from E2–E1 heterodimers to E1 homotrimers [27]. These structural changes introduce variability in compound binding, potentially reducing predictability and stability, as indicated by higher RMSD values and diminished binding efficacy in molecular simulations. This effect has been observed in compounds like CID 25246271, CID 5496862, and CID 3894 [28].

While computational docking and simulation techniques provide valuable insights into these interactions, they may only partially capture the complexity of molecular behavior. These predictions are based on available structural data, which may only account for part of the range of viral mutations or host-specific factors. Like many viruses, CHIKV exhibits genetic variability, which can alter target proteins’ structures and reduce specific inhibitors’ effectiveness. Moreover, host-specific factors such as membrane composition, co-receptor presence, and other cellular components may further influence the efficiency of compound-mediated viral entry inhibition.

Unlike studies that validate their computational results with in vitro or in vivo experiments, this study has not yet advanced to empirical testing. In one study, Puranik et al. (2019) demonstrated that compounds predicted to be highly effective against CHIKV through in silico methods showed varying levels of success in cell-based assays [29]. On the other hand, Ferreira et al. (2010) [30] highlighted that the scoring function used in docking can lead to false positives, often due to the selection of high-energy ligand conformations that appear to fit the target structure well but are energetically difficult for the molecules to adopt. This indicates that, although a compound may receive a promising score in docking simulations, it may not bind effectively in a biological environment [30]. As a result, in vitro validation is essential as it bridges the gap between theoretical predictions and real-world applications. It confirms the computational models’ reliability, assesses the compounds’ practical efficacy and safety, and provides a foundation for further stages of drug development [31].

In summary, while this study offers important insights into potential CHIKV entry inhibitors, future research should integrate these computational findings with experimental validation to fully evaluate the therapeutic potential of the identified compounds against CHIKV and related viruses.

## 4. Materials and Methods

### 4.1. Retrieval and Preparation of Protein Structure

The pre-fusion structure of CHIKV’s mature envelope glycoprotein, along with its native ligand, was obtained from the Protein Data Bank (PDB) (https://www.rcsb.org/) (accessed on 25 June 2023.) [Research Collaboratory for Structural Bioinformatics (RCSB), Piscataway, NJ, USA]. PDB ID of the protein’s X-ray crystal structure is 3N42, with a resolution of 3.0 Å [32]. The structure of the protein was refined using Schrodinger’s Protein Preparation Wizard [Schrödinger, Inc., Schrödinger Suite (Release 2023-2), New York, NY, USA] in order to ensure precise bond ordering, incorporation of hydrogen atoms and absent residues, modification of formal charges in hetero groups, and generation of tautomeric/ionization states at a pH of 7.0 [33], as shown in Figure 6. Water molecules are prevalent in almost all crystallized protein structures [34]. Consequently, water molecules beyond a 5.0 Å radius were removed to focus on the relevant parts of the protein.

### 4.2. Ligand Preparation

Approximately 20,000 potential antiviral compounds were systematically sourced from multiple databases such as ZINC20 [35], Drugbank [36], and Asinex (https://www.asinex.com/antiviral) (accessed on 16 May 2023) and published studies on Chikungunya virus. Bioactive molecules against viruses were obtained from ZINC20, while metabolites, nutraceuticals, and both approved and experimental drugs were selected from DrugBank. Additionally, novel chemical entities with known antiviral activity and improved safety profiles were sourced from Asinex, and compounds previously identified for antiviral activity against CHIKV were selected from PubChem based on supporting literature. These compounds were processed using OpenBabel software version 3.1.1 to format them according to SDF standards, and duplicates were removed using OpenBabel software [University of Pittsburgh, Pittsburgh, PA, USA].

The LigPrep module of the Schrödinger suite was used to optimize and standardize the ligand structures (Schrödinger Release 2023-2). This included generating potential ionization states at physiological pH using the Epik tool [37]. Finally, the compounds were optimized using the OPLS4 force field for accurate virtual screening [38]. From the initial pool of 20,000 compounds, a subset was selected based on molecular properties, drug-likeness, and predicted bioactivity using the QikProp module (Schrödinger Release 2023-2) and Lipinski’s Rule of Five. The QikProp module predicted ADMET properties, excluding compounds that failed to meet Lipinski’s criteria. Furthermore, reactive functional groups were filtered out to minimize potential toxicity or off-target effects. The remaining subset of compounds was then subjected to high-throughput virtual screening.

### 4.3. Receptor Grid Generation

To perform docking analysis, it is necessary to accurately define a grid box on the functional region of the protein. In this molecular docking study, the receptor grid was set up with the Glide grid generation panel [39,40] from the Schrödinger suite. The grid was set up around the amino acid residues on Envelope-2 that interact with the hMXRA8 receptor [12], with an extension to include residues from the Envelope-1 fusion loop [12,18].

The active site was defined by a surrounding grid box measuring 20 × 20 × 20 Å centered on the centroid of the targeted amino acid residues listed in Table 4. A smaller secondary grid box of 10 × 10 × 10 Å was placed within the primary grid to focus on the likely binding site. The outer box dimensions were defined by the coordinates −40.94, −39.31, and −23.99 along the X-, Y-, and Z-axes, respectively, to encompass the active site and accommodate the varying sizes and shapes of potential ligands, as shown in Figure 7. Default parameters were used for the van der Waals radii scaling factor, which was set to 1.0 Å, and a cutoff value of 0.25 was applied for partial atomic charges. No rotatable groups or constraints were specified during the grid generation process.

### 4.4. Molecular Docking

Structure-based screening is an effective method for discovering potential drug candidates [41]. It scans large ligand databases to find molecules that fit and interact with the protein’s active site [42]. Lipinski’s pre-filtering rules and QikProp (Schrödinger Release 2023-2) were also applied. The virtual screening was performed by using the GLIDE module [39,40] in the Schrödinger suite and had three phases: 1. The ligand library was subjected to high-throughput virtual screening (HTVS) to identify the top 40% of molecules based on docking scores. 2. Standard precision (SP) docking was then conducted to identify the top 30% of compounds from the HTVS results. 3. Extra Precision (XP) docking was performed to filter the top 20% of the SP results.

### 4.5. Binding Free Energy Calculations (Rescoring Function)

The compounds from XP results were selected for MM-GBSA (Molecular Mechanics-Generalized Born Surface Area) calculation. We utilized the Prime MM-GBSA method from the Schrödinger suite (Schrödinger Release 2023-2) to calculate the binding free energies of ligand–protein interactions. This method combines molecular mechanics with solvation energies using the OPLS-4 force field [38] and the VSGB solvation model [43]. Docked poses from the Extra Precision (XP) mode of the GLIDE module were processed for this analysis. The binding free energy (ΔG bind) was calculated using the following formula:ΔG bind = E Complex _(minimized)_ − E Ligand _(minimized)_ − E Receptor _(minimized)_

This calculation quantitatively measures the binding affinity, which is crucial for identifying potential lead compounds in our drug discovery process [44].

### 4.6. Ligand-Based ADME/Tox Prediction

Drug-like molecules are characterized by favorable ADMET (absorption, distribution, metabolism, excretion, and toxicity) parameters [45]. We utilized Schrodinger’s QikProp (Schrödinger Release 2023-2) module to predict the drug-likeness and pharmacokinetic profiles of potential Chikungunya virus entry inhibitors. This module compares a molecule’s properties against 95% of known drugs [45,46]. The top compounds, selected based on the docking results and favorable ADMET properties, are displayed in Figure 8.

### 4.7. Molecular Dynamics Simulation and Binding Free Energy Analysis

MD simulation is vital for assessing the stability of protein–ligand complexes [47] Therefore, we conducted MD simulations on the four most promising protein–ligand complexes for 200 ns using Desmond (Schrödinger Release 2023-2) with the OPLS 2005 force field [48]. The simulation system was a cubic box with a 15 Å padding distance from the P-L complex. We used the TIP3P water model for solvation [49]. To mimic physiological conditions, we added 0.15 M NaCl, the temperature kept at 310 K, and the pressure at 1 bar during the simulation period. Ions were included to ensure electrical neutrality.

Finally, the trajectories were analyzed using root-mean-square deviation (RMSD), root-mean-square fluctuation (RMSF), protein–ligand contact mapping and protein–ligand contact timelines to understand the stability of protein–ligand complexes. Since hydrogen bonds are crucial for stable and accurate binding [50,51], we evaluated the proportion of hydrogen bond contacts in the simulation time to understand the occupancy of potential CHIKV entry inhibitors in the receptor binding cavity. Finally, The MM/GBSA analysis was carried out after the simulation using the thermal_MMGBSA.py script from the Prime/Desmond module in the Schrödinger suite 2023-2 [52]. Frames were extracted from the first 100 ns of each MD trajectory, and the binding free energy of the top four compounds was calculated by averaging 500 frames. The Prime MM/GBSA method follows the additivity rule, where the total binding free energy (in kcal/mol) is calculated as the sum of different energy components, including Coulombic, covalent, hydrogen bonding, van der Waals, self-contact, lipophilic, solvation, and π–π stacking interactions between the ligand and the protein.

## 5. Conclusions

This study identifies CID 136801451 as a promising inhibitor of the Chikungunya virus by targeting the E1–E2 glycoprotein complex. With strong binding affinity, favorable pharmacokinetics, and stable interactions in computational analyses, the compound shows potential as an antiviral agent. However, further in vitro and in vivo testing is needed to confirm its effectiveness, which is essential for advancing its development as a therapeutic option for CHIKV.

## Figures and Tables

**Figure 1 ijms-26-06480-f001:**
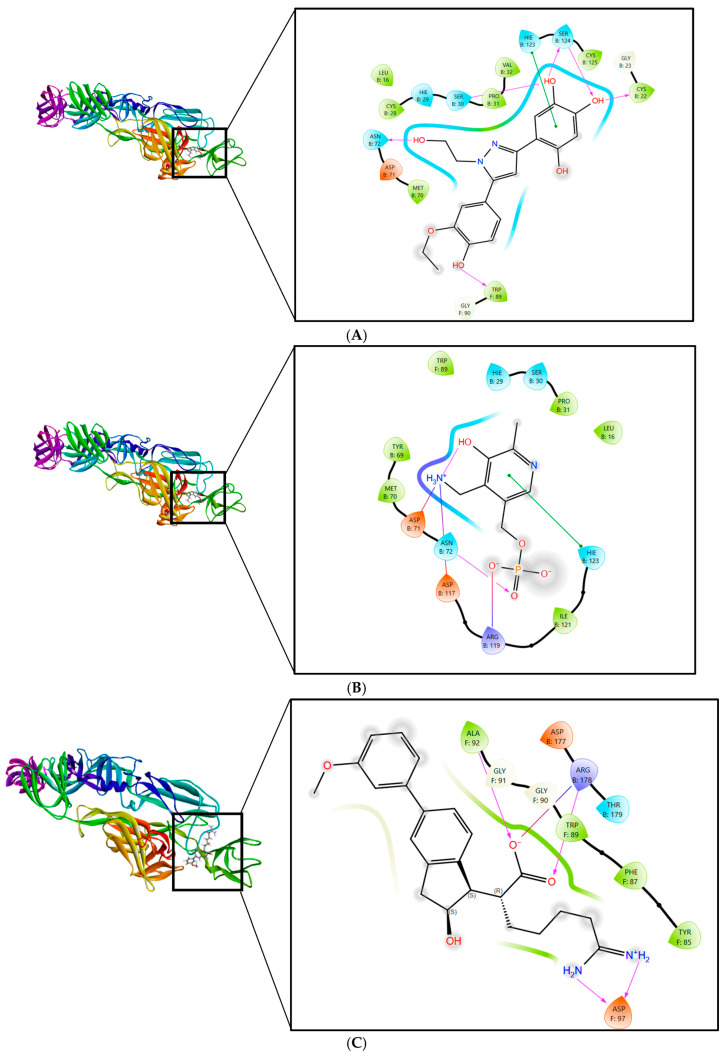
Protein–ligand 2D interaction diagrams—interaction of CHIKV E1–E2 protein complex with (**A**) CID 136801451, (**B**) CID 25246271, (**C**) CID 5496862, (**D**) CID 3894, and (**E**) legend for protein–ligand interaction bond types.

**Figure 2 ijms-26-06480-f002:**
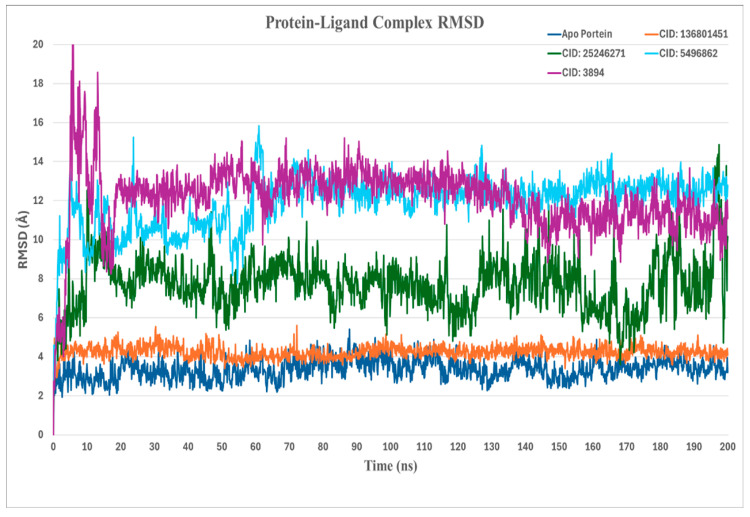
Comparative RMSD trajectories of the four protein–ligand complexes during 200 ns molecular dynamics simulations.

**Figure 3 ijms-26-06480-f003:**
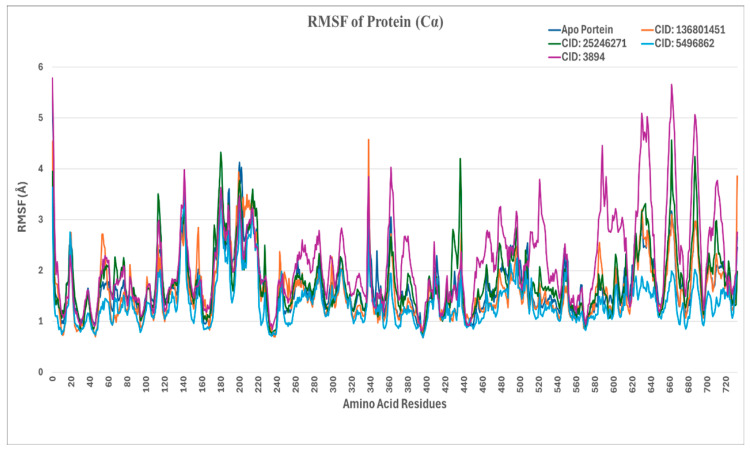
Comparative RMSF trajectories of the four protein–ligand complexes during 200 ns molecular dynamics simulations.

**Figure 4 ijms-26-06480-f004:**
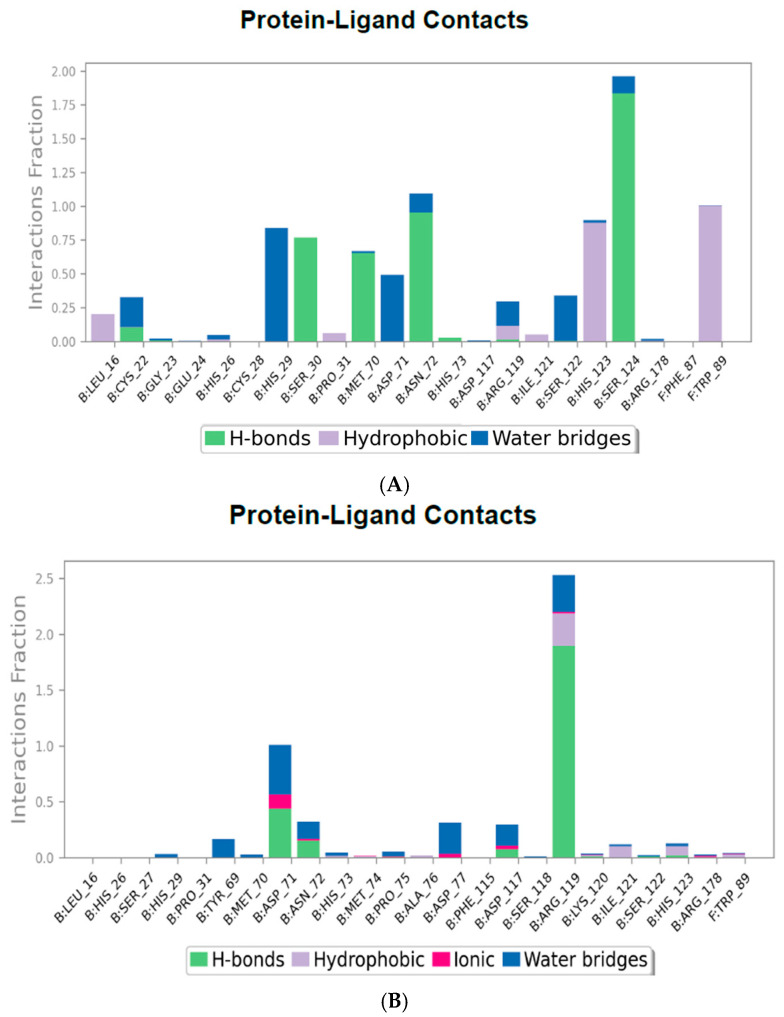
Illustration of protein–ligand contact map of CHIKV envelope E1–E2 protein in complex with the ligands: (**A**) CHIKV envelope E1–E2 in complex with CID 136801451; (**B**) CHIKV envelope E1–E2 in complex with CID 25246271; (**C**) CHIKV envelope E1–E2 in complex with CID 5496862; (**D**) CHIKV envelope E1–E2 in complex with CID 3894.

**Figure 5 ijms-26-06480-f005:**
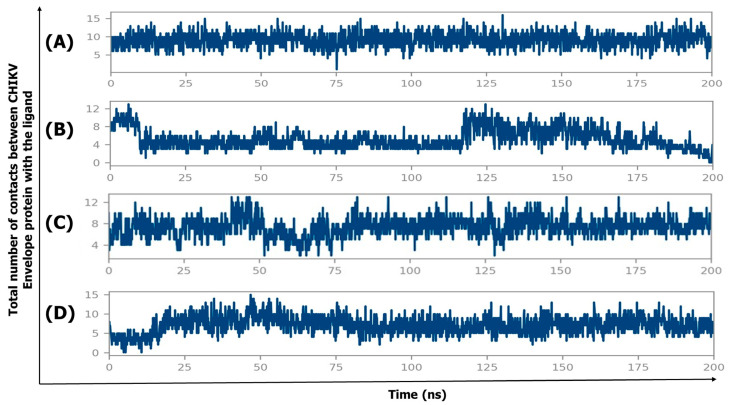
Illustration of protein–ligand contact timeline of CHIKV envelope E1–E2 protein in complex with the ligands: (**A**) CHIKV envelope E1–E2 in complex with CID 136801451; (**B**) CHIKV envelope E1–E2 in complex with CID 25246271; (**C**) CHIKV envelope E1–E2 in complex with CID 5496862; (**D**) CHIKV envelope E1–E2 in complex with CID 3894.

**Figure 6 ijms-26-06480-f006:**
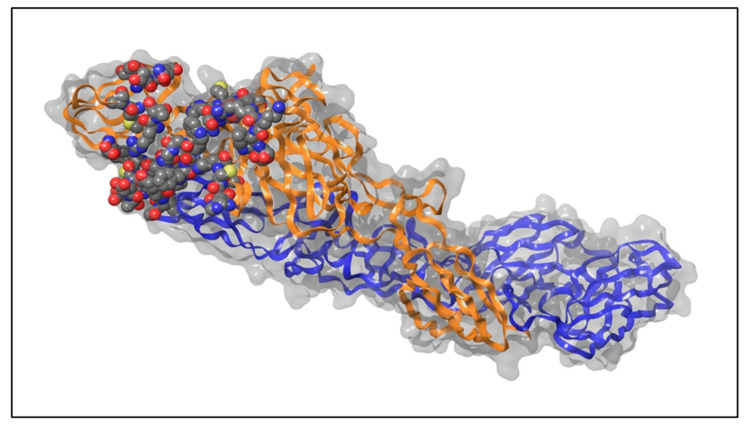
Crystal structure of the mature E2–E1 heterodimeric envelope glycoprotein of Chikungunya virus. E2 is shown as an orange ribbon, E1 is represented as a blue ribbon, and the target site is highlighted using a space-filling molecular model.

**Figure 7 ijms-26-06480-f007:**
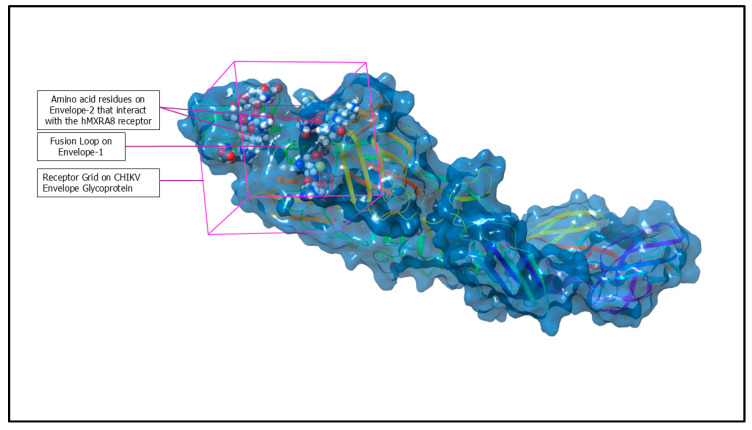
Three-dimensional structure of the CHIKV envelope glycoprotein with receptor grid for docking studies. The highlighted regions include amino acid residues on Envelope-2 (space-filling molecular model) interacting with the hMXRA8 receptor, the fusion loop on Envelope-1 (yellow), and the receptor grid (purple box) used to define the binding site for molecular docking analysis.

**Figure 8 ijms-26-06480-f008:**
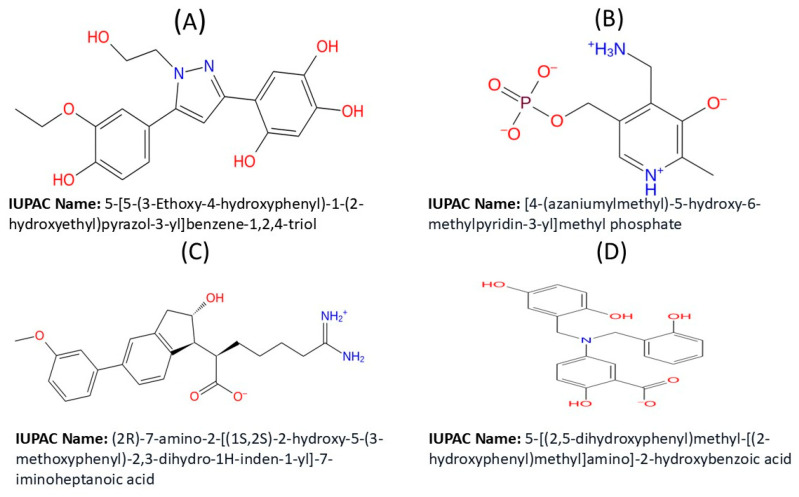
Two-dimensional representation of the selected top compounds: (**A**) CID 136801451; (**B**) CID 25246271; (**C**) CID 5496862; (**D**) CID 3894.

**Table 1 ijms-26-06480-t001:** Results of docking and free energy of binding.

Serial No.	Compound PubChem ID (CID)	XP GScore	Docking Score
1	136801451	−10.227	−10.227
2	25246271	−10.362	−9.39
3	5496862	−9.365	−9.362
4	3894	−9.169	−9.169

**Table 2 ijms-26-06480-t002:** ADMET properties of compounds.

Criteria	CID 136801451	CID 25246271	CID 5496862	CID 3894	Acceptable Range
QPpolrz	36.463	19.185	40.583	34.795	13.0 to 70.0
QPlogPC16	13.309	8.563	14.564	13.336	4.0 to 18.0
QPlogPoct	22.937	18.922	24.058	21.202	8.0 to 35.0
QPlogPw	16.089	16.072	15.246	13.428	4.0 to 45.0
QPlogPo/w	1.764	−2.393	3.336	2.94	−2 to 6.5
QPlogS	−4.176	−0.618	−4.849	−3.281	−6.5 to 0.5
QPlogHERG	−5.794	−0.619	−4.103	−3.269	Concern below −5
QPPCaco	53.22	0.702	17.129	16.864	<25 = poor, >500 = Great
QPlogBB	−2.477	−1.585	−2.519	−2.039	−3 to 12
QPPMDCK	20.768	0.345	7.756	7.627	<25 = poor, >500 = Great
QPlogKp	−4.319	−7.18	−5.971	−3.670	−8 to −1
QPlogKhsa	−0.113	−1.182	0.024	−0.095	−1.5 to 1.5
HOAP	68.166	10.183	68.557	66.117	<25 = poor
#metab	6	7	7	7	1 to 8
#rtvFG	0	1	0	0	0 to 2
CNS	−2	−2	−2	−2	−2 to +2
Glob	0.799196	0.895275	0.793301	34.795	13.0 to 70.0
Rule of Five	0	0	0	0	4.0 to 18.0

QPolrz quantifies predicted polarizability in cubic angstroms, offering insights into molecular electronic properties. QPlogPC16 assesses the predicted hexadecane/gas partition coefficient, indicating hydrophobicity. QPlogPoct estimates the predicted octanol/gas partition coefficient, another hydrophobicity marker. QPlogPw evaluates the predicted water/gas partition coefficient, indicative of aqueous solubility. QlogPo/w determines the predicted octanol/water partition coefficient, a critical measure of lipophilicity. QPlogS projects the predicted aqueous solubility, a key factor in drug absorption and distribution. CIQPlogS offers a conformation-independent solubility prediction, employing a distinct regression equation. QPlogHREG predicts the IC50 value for the blockage of HERG K+ channels, which is crucial for cardiac safety. QPPCaco predicts apparent Caco-2 cell permeability in nm/s, a surrogate for gastrointestinal absorption. QPlogBB estimates the predicted brain/blood partition coefficient relevant for CNS penetration. QPPMDCK assesses predicted MDCK cell permeability in nm/s, indicative of cellular permeability. QPlogKp evaluates predicted skin permeability, significant for transdermal drug delivery. QPlogKhsa forecasts binding affinity to human serum albumin, affecting distribution profiles. HOA (%) predicts human oral absorption on a 0–100% scale, crucial for oral drug viability. #metab quantifies the number of likely metabolic reactions indicative of metabolic stability. #rtvFG counts the number of reactive functional groups pertinent to reactivity and potential toxicity. CNS activity predicts central nervous system activity on a scale from −2 (inactive) to +2 (active). Globularity descriptor (glob) evaluates molecular shape, influencing biological interactions. Rule of Five assesses compliance with Lipinski’s rule for oral drug potential (MW < 500, logPo/w < 5, donorHB < 5, accptHB < 10).

**Table 3 ijms-26-06480-t003:** MM/GBSA-computed binding energies of CHIKV envelope protein–ligand complexes obtained from docking and frames from the molecular dynamics trajectories.

Serial No	Compound PubChem ID (CID)	MMGBSA ΔG Bind Energy of Docked Conformation ^a^ (kcal/mol)	Binding Energy Calculated from Frames of the MD Simulation Trajectories ^b^ (Kcal/mol)
1	136801451	−51.53	−62.48 ± 9.06
2	25246271	−3.58	−8.19 ± 4.84
3	5496862	−20.06	−32.43 ± 11.62
4	3894	−24.18	−38.08 ± 8.40

^a^ Each value corresponds to the single best pose selected from MM/GBSA-optimized conformations in each category of ligand–protein complexes obtained from the GLIDE module in the Schrödinger suite. ^b^ Values represent average ± standard deviation calculated from first 500 frames of 200 ns molecular dynamics simulation.

**Table 4 ijms-26-06480-t004:** Amino acid residues on Envelope-2 and Envelope-1 protein of CHIKV that interact with the hMXRA8 receptor.

CHIKV Envelope	Amino Acids	Positions on Envelope Proteins
Envelope-2 (E2) (Chain-B of 3N42)	His	18, 26, 29
	Ser	27, 182
	Cys	28
	Asp	71, 214, 223
	Asn	72, 193
	Met	74, 181
	Pro	75
	Ala	76
	Arg	119, 178
	Lys	120
	Ile	121
	Thr	179, 191
	Val	192
Envelope-1 (E1) (Chain-F of 3N42)	Gly	83, 90, 91
	Val	84
	Tyr	85, 93
	Pro	86
	Phe	87, 95
	Met	88
	Trp	89
	Ala	92
	Asp	97

## Data Availability

Data will be made available on request.

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
