# Peer review of "Potent Inhibition of Chikungunya Virus Entry by a Pyrazole–Benzene Derivative: A Computational Study Targeting the E1–E2 Glycoprotein Complex"

_ijms, 2025, doi:10.3390/ijms26136480_

Round 1
Reviewer 1 Report
Comments and Suggestions for Authors
The manuscript by Rahman and colleagues presents a computational investigation into potential inhibitors of Chikungunya virus entry, focusing on the E1–E2 glycoprotein complex. The integration of high-throughput virtual screening, binding free energy estimation, ADMET analysis, and extended molecular dynamics simulations provides a pipeline for candidate identification, and the identification of CID 136801451 as a lead compound with strong binding affinity and stable interactions is promising. However, the manuscript would benefit from a clearer discussion on the selection criteria for the initial compound library and a more detailed comparison of CID 136801451 with known CHIKV inhibitors, if available. Additionally, while the computational results are compelling, the lack of experimental validation limits the immediate translational impact. Overall, the study offers valuable insights and provides a strong foundation for subsequent experimental investigations. Other points of the manuscript that require revision are listed below:
- Line 98: Figure 3 was mistakenly cited in place of Figure 1.
- Line 116: Data in Table 2 requires a proper analysis in section 2.3.
- Line 191: Figure 4 requires improvement in resolution to allow reading.
- Lines 197/200/203/205: Figure 6 was mistakenly cited in place of Figure 4.
- Line 276: No supplementary material was provided with the manuscript.
- Lines 326-328: Figure 5 is not mentioned in the main text of the manuscript.
- Lines 330-332: It is not clear how potential antiviral compounds were sourced from databases.
- Lines 340-343: References for the amino acids on E2 protein known to interact with the MXRA8 receptor and those on the E1 protein fusion loop are missing.
- Line 347: Table 1 was mistakenly cited in place of Table 3.
Author Response
Comments 1: Line 98: Figure 3 was mistakenly cited in place of Figure 1. |
Response 1: Thank you for pointing this out. The error has been corrected. Figure 1 is now properly cited in place of Figure 3. This change can be found in the revised manuscript on page 4, line 102. |
Comments 2: Line 116: Data in Table 2 requires a proper analysis in section 2.3. |
Response 2: A detailed analysis of the data from Table 2 has been added in section 2.3. This change can be found in the revised manuscript on page 6, lines 115-130.
“[updated text in the manuscript 2.3. ADME/T Properties Analysis The ADME (Absorption, Distribution, Metabolism, and Excretion) and Toxicity profiles of compounds identified as potential inhibitors of the Chikungunya virus were systematically evaluated and presented in Table 2. CID 136801451 stands out as the most promising compound, exhibiting strong oral absorption, favorable permeability (QPP-Caco: 53.22), and minimal central nervous system (CNS) activity, making it an ideal candidate for further development. It also demonstrates a good balance of lipophilicity and solubility, with a QPlogPC16 of 13.309 and a QPlogS of -4.176, although it raises slight concern for cardiotoxicity (QPlogHERG: -5.794). CID 5496862 shows favorable oral absorption and solubility, with a QPlogPC16 of 14.564 and a QPlogS of -4.849, but its permeability (QPPCaco: 17.129) and overall bioavailability are lower than CID 136801451. CID 3894 offers better skin permeability (QPlogKp: -3.670) and moderate metabolic stability (7 reactions), but its permeability and solubility are still lower com-pared to CID 136801451, making it less suitable for oral bioavailability. CID 25246271, however, shows poor performance across multiple parameters, including low solubility (QPlogS: -0.618), permeability (QPPCaco: 0.702), and human oral absorption (HOAP: 10.183), indicating it is less favorable for oral administration and overall efficacy. Despite these differences, all compounds comply with the Rule of Five, indicating they generally possess favorable drug-like properties.]” |
Comments 3: Line 191: Figure 4 requires improvement in resolution to allow reading. |
Response 3: Thank you for your feedback. The resolution of Figure 4 has been improved, and the layout has been adjusted to enhance readability. This change is reflected in page and line number Page 10/11.
|
Comments 4: Lines 197/200/203/205: Figure 6 was mistakenly cited in place of Figure 4. |
Response 4: Thank you for pointing out this error. The citations have been corrected, and Figure 4 is now properly cited in place of Figure 6. This change can be found in the revised manuscript on page 11, line 217/222/225/228/230
|
Comments 5: Line 276: No supplementary material was provided with the manuscript. |
Response 5: The ADMET properties of the compounds are now thoroughly analyzed in section 2.3 (page 6, lines 115-130). As a result, no supplementary material is required anymore.
|
|
Comments 6: Lines 326-328: Figure 5 is not mentioned in the main text of the manuscript. |
|
Response 6: Figure 5 has been changed to Figure 6 and is now referenced in the main text of the manuscript. This update can be found on page 16, line 434 and page 16, line 438.
|
Comments 7: Lines 330-332: It is not clear how potential antiviral compounds were sourced from databases. |
|
Response 7: Information on the 20,000 compound library composition has been included in the revised manuscript. This can be found on page 17, lines 442-461.
[Updated Text in the Manuscript: 4.2. Ligand Preparation Approximately 20,000 potential antiviral compounds were systematically sourced from multiple databases such as ZINC20 [35], Drugbank [36], Asinex (https://www.asinex.com/antiviral) and published studies on Chikungunya virus. Bioactive molecules against viruses were obtained from ZINC20, while metabolites, nutraceuticals, and both approved and experimental drugs were selected from DrugBank. Additionally, novel chemical entities with known antiviral activity and improved safety profiles were sourced from Asinex, and compounds previously identified for antiviral activity against CHIKV were selected from PubChem based on supporting literature. These compounds were processed using OpenBabel to format them according to SDF standards, and duplicates were removed using OpenBabel software. The LigPrep module of the Schrödinger suite was used to optimize and standardize the ligand structures (Schrödinger Release 2023-2). This included generating potential ionization states at physiological pH using the Epik tool [37]. Finally, the compounds were optimized using the OPLS4 force field for accurate virtual screening [38]. From the initial pool of 20,000 compounds, a subset was selected based on molecular properties, drug-likeness, and predicted bioactivity using the QikProp module (Schrödinger Release 2023-2) and Lipinski’s Rule of Five. The QikProp module predicted ADMET properties, excluding compounds that failed to meet Lipinski's criteria. Furthermore, reactive functional groups were filtered out to minimize potential toxicity or off-target effects. The remaining subset of compounds was then subjected to high-throughput virtual screening.]
|
|
Comments 8: Lines 340-343: References for the amino acids on E2 protein known to interact with the MXRA8 receptor and those on the E1 protein fusion loop are missing. |
|
Response 8: References for the amino acids on the E2 protein and the E1 protein fusion loop have been added. This can be found in page 17, lines 467-468.
[Updated Text in the Manuscript: 4.3. Receptor Grid Generation To perform docking analysis, it is necessary to accurately define a grid box on the functional region of the protein. In this molecular docking study, the receptor grid was set up with the Glide grid generation panel [39], [40] from the Schrödinger suite. The grid was set up around the amino acid residues on Envelope-2 that interact with the hMXRA8 receptor [12], with an extension to include residues from the Envelope-1 fusion loop [12], [18].]
|
|
Comments 9: Line 347: Table 1 was mistakenly cited in place of Table 3 |
|
Response 9: Thank you for pointing this out. The citation has been corrected. Table 3 is now changed to Table 4 and properly cited in the manuscript. This change can be found in the revised manuscript on page 18, line 472.
|
|
|
|
Reviewer 2 Report
Comments and Suggestions for Authors
The article “Potent Inhibition of Chikungunya Virus Entry by a Pyrazole-Benzene Derivative: A Computational Study Targeting the E1-E2 Glycoprotein Complex” presents a virtual screening approach to identify potential Chikungunya virus (CHIKV) entry inhibitors targeting the E1-E2 glycoprotein complex. However, the manuscript has several limitations that need to be addressed before publication.
Major limitations:
- This is purely a computational study without any experimental validation. The identified compound (CID 136801451) requires in vitro testing to confirm predicted activity. But what is more important: No comparison with known CHIKV inhibitors in terms of binding affinity or predicted efficacy were carried.
- There are some methodology issues: The receptor grid generation lacks sufficient detail about the specific binding site rationale.
- Information on the 20,000 compound library composition and filtering criteria must be extended and explained the rationale behind chosing it.
- The dramatic difference in ΔG_bind values between compounds (e.g., CID 136801451: -51.53 vs CID 25246271: -3.58 kcal/mol) seems unusually large and may indicate calculation errors. Furthermore, no statistical significance testing of docking scores or binding energies. The absence of positive controls or known inhibitors for comparative analysis might be behind those issues.
- The discussion on why some compounds showed higher RMSD values needs to be further explored and extended.
- The authors need to critically analyze the computational limitations related to carrying an all-computational study, e.g., discussing the potential false positives inherent in virtual screening.
Minor issues:
- Figure 2 and 3 labels are inconsistent (referenced as Figure 2 and 3, but caption mentions Figure 3)
- Figures suffer from low quality and distortion, the authors need to fix these issues in order to achieve sufficient publication quality.
The manuscript needs substantial improvements in presentation, analysis and critical evaluation of the computational limitations before it can be considered for publication.
Author Response
Comment 1:
This is purely a computational study without any experimental validation. The identified compound (CID 136801451) requires in vitro testing to confirm predicted activity. But what is more important: No comparison with known CHIKV inhibitors in terms of binding affinity or predicted efficacy were carried.
Response 1:
We agree with this comment. We are currently working to test the identified CHIKV entry inhibitor (CID 136801451) in vitro and encourage other researchers to validate its efficacy as well. Regarding the comparison with known CHIKV inhibitors, we included all compounds reported to have anti-CHIKV activity from the literature in our compound library. However, from the virtual screening results, none of these compounds demonstrated promising interactions with our target site, as we focused on a unique region of the CHIKV E2-E1 glycoprotein. This site encompasses the hMXRA8 interacting region and the fusion loop, as described in section 4.3. Our dual-targeting strategy aims to block CHIKV entry into host cells in two steps: first, by preventing the interaction between the envelope and the host cell receptor, and second, by inhibiting membrane fusion through the fusion loop. Since no existing compound targets this site and no other compound showed significant interactions, CID 136801451 remains the most promising candidate. This finding is discussed in the manuscript on Page 15, lines 380-383.
Comment 2:
There are some methodology issues: The receptor grid generation lacks sufficient detail about the specific binding site rationale.
Response 2:
Thank you for pointing this out. We have revised section 4.3 (Receptor Grid Generation) on pages 17-19, lines 466-489. References have been added to support the selection of amino acid residues in the grid. The coordinates for the X, Y, and Z axes of the grid are now provided. The receptor grid generation process has been explained with greater clarity. Additionally, a new figure (Figure 7) has been added to illustrate the receptor grid generation process for better understanding.
Comment 3:
Information on the 20,000 compound library composition and filtering criteria must be extended and explained the rationale behind choosing it.
Response 3:
Agree. We have modified section 4.2 (Ligand Preparation) to address this point in more detail.
- Information on the 20,000 compound library composition has been included on page 17, lines 447-455.
- The filtering criteria for the compound library have been added on page 17, lines 459-465.
[Updated Text in the Manuscript:
4.2. Ligand Preparation
Approximately 20,000 potential antiviral compounds were systematically sourced from multiple databases such as ZINC20 [35], Drugbank [36], Asinex (https://www.asinex.com/antiviral) and published studies on Chikungunya virus. Bioactive molecules against viruses were obtained from ZINC20, while metabolites, nutraceuticals, and both approved and experimental drugs were selected from DrugBank. Additionally, novel chemical entities with known antiviral activity and improved safety profiles were sourced from Asinex, and compounds previously identified for antiviral activity against CHIKV were selected from PubChem based on supporting literature. These compounds were processed using OpenBabel to format them according to SDF standards, and duplicates were removed using OpenBabel software.
The LigPrep module of the Schrödinger suite was used to optimize and standardize the ligand structures (Schrödinger Release 2023-2). This included generating potential ionization states at physiological pH using the Epik tool [37]. Finally, the compounds were optimized using the OPLS4 force field for accurate virtual screening [38]. From the initial pool of 20,000 compounds, a subset was selected based on molecular properties, drug-likeness, and predicted bioactivity using the QikProp module (Schrödinger Release 2023-2) and Lipinski’s Rule of Five. The QikProp module predicted ADMET properties, excluding compounds that failed to meet Lipinski's criteria. Furthermore, reactive functional groups were filtered out to minimize potential toxicity or off-target effects. The remaining subset of compounds was then subjected to high-throughput virtual screening.]
Comment 4:
The dramatic difference in ΔG_bind values between compounds (e.g., CID 136801451: -51.53 vs CID 25246271: -3.58 kcal/mol) seems unusually large and may indicate calculation errors. Furthermore, no statistical significance testing of docking scores or binding energies. The absence of positive controls or known inhibitors for comparative analysis might be behind those issues.
Response 4:
We have conducted post-simulation MM/GBSA analysis from molecular dynamics (MD) simulation trajectories to statistically validate the binding energies obtained from virtual screening. The results are presented on page 12, lines 272-294, in section 2.4.5 of the manuscript. Additionally, the Molecular Dynamics Simulation methodology has been updated on page 20, lines 535-543 to provide more clarity.
The dramatic difference in ΔG_bind values between compounds arises from fundamental differences in their molecular interactions and the scoring methods used to account for binding energetics. CID 136801451 forms extensive hydrogen bonds and hydrophobic interactions with both the E1 and E2 domains of the CHIKV envelope protein, targeting critical residues involved in viral fusion and receptor binding. In contrast, CID 25246271 forms minimal hydrogen bonds and weak hydrophobic interactions.
While the XP docking stage was used to prioritize top candidates due to its accuracy and efficiency, it is limited to a static view of rigid receptor-ligand interactions and often overestimates the affinity of ligands that fit geometrically but lack dynamic stability. Docking scores (e.g., Glide XP) primarily estimate enthalpic contributions without fully accounting for entropic factors, solvation effects, or receptor flexibility. Consequently, CID 25246271 showed a favorable docking score (–9.39) despite its weak dynamic stability. Post-docking MM/GBSA calculations from MD simulations, which incorporate protein flexibility and solvation, provided a more realistic binding energy for CID 25246271 (–8.19 ± 4.84 kcal/mol). The MD simulation further revealed high RMSD values and transient protein-ligand contacts for CID 25246271, confirming its unstable binding.
In contrast, CID 136801451 maintained consistent, strong interactions throughout the simulation, supporting its significantly stronger ΔG_bind values and higher inhibitor potential. These findings underscore the importance of integrating both docking and dynamic simulations for more accurate affinity predictions.
Regarding the positive controls, we have included all compounds reported to have anti-CHIKV activity in the literature, which were added to our compound library. However, as we targeted a unique site on the CHIKV E2-E1 glycoprotein, none of the compounds showed promising interactions with our target site.
[Updated Text in the Manuscript:
2.4.5. MM/GBSA computations from MD trajectories (From Result Section)
Post-simulation MM/GBSA calculations were performed using frames 1–500 from the MD simulation trajectories, totaling 500 conformations for each simulated complex. The average binding energies, along with their standard deviations, are presented in Table 3. MM/GBSA is a widely used and rigorous method for post-simulation binding free energy prediction because it accounts for factors such as protein flexibility, entropy, solvation, and polarizability—factors often overlooked in many docking protocols. Among the four CHIKV Envelope Protein-Ligand complexes, CID 136801451 exhibits the strongest binding affinity, with a docking energy of -51.53 kcal/mol and a significantly improved MD simulation binding energy of -62.48 ± 9.06 kcal/mol, indicating strong and stable interactions. CID 3894 shows a docking energy of -24.18 kcal/mol and an enhanced MD energy of -38.08 ± 8.40 kcal/mol, suggesting it also stabilizes over time. CID 5496862 demonstrates moderate binding, with a docking energy of -20.06 kcal/mol and a stronger MD energy of -32.43 ± 11.62 kcal/mol, indicating an improvement in binding during the simulation. Conversely, CID 25246271 shows the weakest binding in both docking (-3.58 kcal/mol) and MD simulation (-8.19 ± 4.84 kcal/mol).
Table 3. MM/GBSA-computed binding energies of CHIKV Envelope Protein-Ligand complexes obtained from docking and frames from the molecular dynamics trajectories.
Serial No |
Compound PubChem ID (CID) |
MMGBSA ΔG Bind Energy of Docked Conformationa (kcal/mol) |
Binding Energy Calculated from Frames of the MD Simulation Trajectoriesb (Kcal/mol) |
|
1 |
136801451 |
-51.53 |
-62.48 ± 9.06 |
|
2 |
25246271 |
-3.58 |
-8.19 ± 4.84 |
|
3 |
5496862 |
-20.06 |
-32.43 ± 11.62 |
|
4 |
3894 |
-24.18 |
-38.08 ± 8.40 |
a Each value corresponds to single best pose selected from MM/GBSA-optimized conformations in each category of ligand-protein complexes obtained from GLIDE module in the Schrödinger suite.
b Values represent average ± standard deviation calculated from first 500 frames of 200 ns molecular dynamics simulation.
4.7. Molecular Dynamics Simulation (From Methodology Section)
Finally, The MM/GBSA analysis was carried out after the simulation using the thermal_MMGBSA.py script from the Prime/Desmond module in the Schrödinger suite 2023-2 [52]. Frames were extracted from the first 100 ns of each MD trajectory, and the binding free energy of the top four compounds was calculated by averaging 500 frames. The Prime MM/GBSA method follows the additivity rule, where the total binding free energy (in kcal/mol) is calculated as the sum of different energy components, including Coulombic, covalent, hydrogen bonding, van der Waals, self-contact, lipophilic, solvation, and π-π stacking interactions between the ligand and the protein.]
Comment 5:
The discussion on why some compounds showed higher RMSD values needs to be further explored and extended.
Response 5:
To better understand the reasons behind the higher RMSD values of some compounds, we have added a new analysis in the results section (2.4.4. Protein-Ligand Contact Timeline) on page 11, lines 239-270. Additionally, the discussion on why some compounds showed higher RMSD values has been extended in the manuscript on pages 14-15, lines 320-352.
[Updated Text in the Manuscript:
2.4.4. Protein-Ligand Contact Timeline
The protein-ligand contact timelines provide valuable insights into the binding stability and interaction dynamics of the ligands with the CHIKV envelope E1–E2 protein. These contacts represent the various types of interactions observed during the simulation, including hydrogen bonds, hydrophobic interactions, ionic contacts, and water bridges. Figure 5 shows the protein-ligand contact timelines for the CHIKV envelope E1–E2 protein in complex with four different ligands (CID 136801451, CID 25246271, CID 5496862, and CID 3894) over a 200 ns molecular dynamics (MD) simulation. The y-axis represents the total number of contacts between the ligand and the protein, while the x-axis represents the simulation time in nanoseconds (ns).
Figure 5. Illustration of protein-ligand contact timeline of CHIKV envelope E1–E2 protein in Complex with the ligands (A) CHIKV envelope E1–E2 in complex with CID 136801451, (B) CHIKV envelope E1–E2 in complex with CID 25246271, (C) CHIKV envelope E1–E2 in complex with CID 5496862, (D) CHIKV envelope E1–E2 in complex with CID 3894.
CID 136801451 (Panel A) demonstrates the most stable and consistent interaction throughout the simulation. The contact numbers remain relatively constant (around 10) over time, indicating that this ligand maintains a strong and persistent binding with the CHIKV envelope protein throughout the entire simulation. In contrast, CID 25246271 (Panel B) exhibits a more transient and unstable binding pattern. The contact numbers fluctuate between 0 and 12, with several periods where the contact count drops to near zero. These fluctuations suggest that CID 25246271 fails to maintain consistent contact with the protein, likely forming weaker or less stable binding interactions.
CID 5496862 (Panel C) shows a similar fluctuating contact pattern to CID 25246271, with contact numbers ranging between 4 and 12. Although the contact numbers are higher compared to CID 25246271, they are still less stable and lower than those observed with CID 136801451. This indicates that CID 5496862 has low to moderate binding stability with the CHIKV envelope protein. CID 3894 (Panel D) initially shows few contacts with the protein, but after approximately 25 ns, the ligand maintains contact numbers ranging from 5 to 10 for the remainder of the simulation. This suggests that CID 3894 forms a moderately stable protein-ligand interaction after an initial period of weaker binding.
Revised Discussion part line pages 14-15, lines 320-352.:
Molecular dynamics simulations further confirmed the stability of CID 136801451 within the E1-E2 complex. Throughout a 200-nanosecond simulation, the compound maintained a consistent root-mean-square deviation (RMSD) of approximately 4 Å. This low RMSD was supported by around 10 stable protein-ligand (P-L) contacts, as shown in Figure 5A. In contrast, other compounds, such as CID 25246271, formed fewer P-L contacts (Figure 5B) and exhibited greater fluctuations in RMSD during the MD simulation, indicating lower stability.
CID 5496862 initially maintained very few P-L contacts (up to 75 ns) with several drops in contact numbers (Figure 5C). This led to an increase in RMSD, reaching up to 16 Å. However, after 75 ns, CID 5496862 stabilized, maintaining nearly 8 P-L contacts with the target protein, resulting in reduced RMSD fluctuations, with the ligand showing a stable RMSD value around 13 Å for the remainder of the simulation. Similarly, CID 3894 exhibited a dramatic increase in RMSD, rising from 0 to 20 Å within the first 10 ns of the simulation. During the first 25 ns, the compound experienced significant RMSD fluctuations due to a low number of P-L contacts (Figure 5D). As the simulation progressed, CID 3894 moved from its initial reference position—defined as the position obtained from protein-ligand docking during virtual screening—to a new conformation, where it formed more stable interactions, including hydrogen bonds, hydrophobic interactions, ionic bonds, and water bridges. This transition led to a more stable interaction with the CHIKV envelope protein, supported by reduced RMSD fluctuations after 25 ns.
Both CID 5496862 and CID 3894 initially exhibited significantly high RMSD values due to the movement of the ligands from their reference positions (obtained through docking in virtual screening) to new conformations. This movement resulted in inefficient interactions with the target amino acid residues (Table 3) of the CHIKV E2-E1 protein complex, as evidenced by the protein-ligand contact maps (Figure 4), indicating lower effectiveness as CHIKV entry inhibitors compared to CID 136801451.]
Comment 6:
The authors need to critically analyze the computational limitations related to carrying out an all-computational study, e.g., discussing the potential false positives inherent in virtual screening.
Response 6:
We have added an extended discussion on the computational limitations of conducting an all-computational study, including the potential for false positives in virtual screening.
This is discussed on page 16, lines 414-422.
[Updated text in Manuscript:
While computational docking and simulation techniques provide valuable insights into these interactions, they may only partially capture the complexity of molecular behavior. These predictions are based on available structural data, which may only account for part of the range of viral mutations or host-specific factors. Like many viruses, CHIKV exhibits genetic variability, which can alter target proteins' structures and reduce specific inhibitors' effectiveness. Moreover, host-specific factors such as membrane composition, co-receptor presence, and other cellular components may further influence the efficiency of compound-mediated viral entry inhibition.
Unlike studies that validate their computational results with in vitro or in vivo experiments, this study has not yet advanced to empirical testing. In one study, Puranik et al. (2019) demonstrated that compounds predicted to be highly effective against CHIKV through in silico methods showed varying levels of success in cell-based assays [30]. On the other hand, Ferreira et al. (2010) highlighted that the scoring function used in docking can lead to false positives, often due to the selection of high-energy ligand conformations that appear to fit the target structure well but are energetically difficult for the molecules to adopt. This indicates that, although a compound may receive a promising score in docking simulations, it may not actually bind effectively in a biological environment [52]. As a result, in-vitro validation is essential as it bridges the gap between theoretical predictions and real-world applications. It confirms the computational models' reliability, assesses the compounds' practical efficacy and safety, and provides a foundation for further stages of drug development [31].]
Comment 7:
Figure 2 and 3 labels are inconsistent (referenced as Figure 2 and 3, but caption mentions Figure 3)
Thank you for noting this inconsistency. The labels and captions for Figures 2 and 3 have been corrected.
- Figure 2: page 8, lines 164 and 169
- Figure 3: page 9, lines 184 and 187
Comment 8:
Figures suffer from low quality and distortion, the authors need to fix these issues in order to achieve sufficient publication quality.
Response 8:
We have replaced the original images with higher-resolution versions, adjusted the aspect ratios, and refined the layouts to enhance clarity.
- Figure 1: updated on page 5/6, line 113/114
- Figure 4: updated on page 10/11, line 213/222

Round 2
Reviewer 2 Report
Comments and Suggestions for Authors
The authors have succesfully answered the questions